# The Ang-(1–7)/MasR axis ameliorates neuroinflammation in hypothermic traumatic brain injury in mice by modulating phenotypic transformation of microglia

Dan Ye[1]☯, Jiamin Liu[2]☯, Long Lin[3], Pengwei Hou[3], Tianshun Feng[4], Shousen Wang[5]*

1 Department of Neurosurgery, Fuzong Teaching Hospital, Fujian University of Traditional Chinese Medicine, Fuzhou, Fujian, China, 2 College of Integrative Medicine, Fujian University of Traditional Chinese Medicine, Fuzhou, Fujian, China, 3 Department of Neurosurgery, Fuzong Clinical Medical College, Fujian Medical University, Fuzhou, Fujian, China, 4 Department of Neurosurgery, Dongfang Affiliated Hospital of Xiamen University, School of Medicine, Xiamen University, Xiamen, China, 5 Department of Neurosurgery, 900th Hospital, Fuzhou, Fujian, China

☯ These authors contributed equally to this work.
* wshsen1965@126.com

**Data Availability Statement:** All relevant data are within the paper and its Supporting Information files.

## Abstract

The Ang-(1–7)/MasR axis is critically involved in treating several diseases; For example, Ang-(1–7) improves inflammatory response and neurological function after traumatic brain injury and inhibits post-inflammatory hypothermia. However, its function in traumatic brain injury (TBI) combined with seawater immersion hypothermia remains unclear. Here, we used a mice model of hypothermic TBI and a BV2 cell model of hypothermic inflammation to investigate whether the Ang-(1–7)/MasR axis is involved in ameliorating hypothermic TBI. Quantitative reverse transcription PCR, western blotting assay, and immunofluorescence assay were performed to confirm microglia polarization and cytokine regulation. Hematoxylin-eosin staining, Nissl staining, and immunohistochemical assay were conducted to assess the extent of hypothermic TBI-induced damage and the ameliorative effect of Ang-(1–7) in mice. An open field experiment and neurological function scoring with two approaches were used to assess the degree of recovery and prognosis in mice. After hypothermic TBI establishment in BV2 cells, the Ang-(1–7)/MasR axis induced phenotypic transformation of microglia from M1 to M2, inhibited IL-6 and IL-1β release, and upregulated IL-4 and IL-10 levels. After hypothermic TBI development in mice, intraperitoneally administered Ang-(1–7) attenuated histological damage and promoted neurological recovery. These findings suggest that hypothermia exacerbates TBI-induced damage and that the Ang-(1–7)/MasR axis can ameliorate hypothermic TBI and directly affect prognosis.

## Introduction

Individuals with traumatic brain injury (TBI) are highly susceptible to develop comorbid hypothermia if they fall into the sea during a maritime accident. Severe hypothermia is one of

**Funding:** This work was supported by the 900th Hospital of Commanding Project and Special Treatment for Trauma (grant no. 2022ZL01 and 2022ZL03), the Projects of LQZD-SW and Fujian Provincial Science and Technology Programme Science and Technology Innovation Platform Project (grant no. 2022Y2017).

**Competing interests:** The authors declare no conflict of interest.

the most prominent causes of death in marine accidents. It is estimated that approximately 700,000 people globally suffer from TBI each year [1], and nearly 375,000 individuals die due to drowning in seawater [2]. However, there are no appropriate methods to reduce the mortality rate and improve the prognosis of individuals affected by TBI-SIH.Therefore, it is important to study the pathophysiological mechanisms associated with the development of TBI combined with seawater immersion hypothermia (TBI-SIH).

Previous studies have shown that following TBI, neuroinflammation develops and persists as the most important condition secondary to brain injury, thereby severely affecting patient prognosis [3,4]. Microglia are considered key players in the development of neuroinflammation, and it is currently believed that M1 phenotype microglia tend to release destructive factors such as interleukin-6 (IL-6), IL-1β, and inducible nitric oxide synthase, whereas M2 phenotype microglia release factors that promote inflammation reduction and tissue healing [5,6].

Angiotensin-(1–7) [Ang-(1–7)] is considered a degradation product of Ang I and Ang II; it binds to the G-protein-coupled receptor Mas (MasR) and antagonizes the effects of Ang II [7]. Ang-(1–7) exerts beneficial effects on the cardiovascular system and shows other metabolic effects such as anti-hypertensive effect and amelioration of organ damage [8–10]. Recent studies have shown that the Ang-(1–7)/MasR axis is a protective arm of the renin-angiotensin system against inflammation and improves the prognosis of TBI [11,12]. Guangjin Gu et al. suggested that Ang-(1–7) could regulate microglia polarization from M1 to M2 phenotype after spinal cord injury, thereby promoting functional recovery [13].

Currently, although several treatment methods for TBI have been developed, there are no studies reporting therapeutic approaches for treating TBI-SIH. Hence, the present study aimed to determine whether the Ang-(1–7)/MasR axis exerts a therapeutic effect on hypothermic TBI and the underlying mechanisms; the findings of this study could guide to develop appropriate treatment methods for TBI-SIH.

## Materials and methods

### Animal preparation

The animal experiment was approved by the Animal Care and Use Committee of Fujian University of Traditional Chinese Medicine and complied with the Guide for the Care and Use of Laboratory Animals (no. 2020–051; National Research Council Institute for Laboratory Animal Research, Inc., Washington, DC: National Academy Press, 1996). For the present study, C57BL/6 male mice weighing 25–28 g were obtained from the Animal Experimentation Centre of the 900th Hospital, Fuzhou, China. The animals were maintained in a standard environment under a 12-h light/dark cycle, with ad libitum access to food and water.

### Ethics approval

The animal experiment was approved by the Animal Care and Use Committee of the 900th Hospital and complied with the Guide for the Care and Use of Laboratory Animals (no. 2024–02; National Research Council Institute for Laboratory Animal Research, Inc., Washington, DC: National Academy Press, 1996).

### Development of the TBI-SIH animal model and drug treatment

All mice were fasted and dehydrated before surgery and then assigned to four groups: control group, TBI combined with 15˚C seawater immersion (TS) group, TS+Ang-(1–7) group, and TS+Ang-(1–7)+A779 group. The mice in the TS group were anesthetized using a small animal

anesthesia machine (RWD Life Science Co., Shenzhen, China). Following anesthesia, TBI was induced in mice by using a CCI device. The mice were fixed on a stereotactic apparatus and subjected to a 3-mm craniectomy. The exposed brain surface was impacted with a 2-mm metal flat-tip impactor (RWD Life Science Co.) at the speed of 4 m/s, a depth of 1.5 mm, and an impact dwell time of 0.1 s. After the wound was hemostatically sutured, the mice were immobilized and allowed to breath freely in a homemade seawater immersion device. Once the animals regained consciousness (appearance of corneal reflexes and tingling responses), they were placed in a thermostatic tank (Jiangsu Hengmin Instrument Manufacturing Co., Ltd., Jiangsu, China) containing natural seawater (Fujian, China); the depth of submersion was the clavicle level, and the duration of submersion was 1 h. Ang-(1–7) (2.0 mg/kg) and A779 (5 mg/kg) were administered intraperitoneally after 2 h. For comparison, the control and TS groups were administered intraperitoneally with equal doses of PBS.

## BV2 cell culture and model establishment

Mouse microglia BV2 cells were purchased from Wuhan Punosai Life Science and Technology Co., Ltd., China. The cells were grown in minimal essential medium (containing non-essential amino acids) (Wuhan Punosai Life Science and Technology Co., Ltd.) supplemented with 10% fetal bovine serum and 1% penicillin/streptomycin and cultured in a humidified incubator at 37°C and 5% $CO_2$.

BV2 cells in the exponential growth phase were assigned to four groups: control group, lipopolysaccharide (LPS) + low-temperature-induced (TS) group, TS+Ang-(1–7) group, and TS+Ang-(1–7)+A7779 group. TS induction was performed at 15°C for 1 h after LPS addition to the cultured cells. The cells were pre-treated with A779, a Mas receptor antagonist, 30 min before LPS addition and Ang-(1–7) intervention. BV2 cells were treated with LPS (1 μg/mL, L3025, Sigma), Ang-(1–7) ($10^{-6}$ mol/L, HY-12403, MCE), and A779 ($10^{-5}$ mol/L, HY-P0216, MCE) for 24 h.

## RNA extraction and quantitative reverse transcription-PCR

Total RNA was extracted from BV2 cells by using an extraction buffer (TRIzol/phenol/chloroform) and quantified by spectrophotometric analysis ($OD_{260}$/$OD_{280}$). After reverse transcription of total mRNA to cDNA, real-time quantitative PCR (RT-qPCR) was performed using SYBR Green PCR Master Mix (Novozymes Biotechnology Ltd., CW 0957). The primers used for the experiment were as follows:

Gene Forward primer sequence (5′-3′) Reverse primer sequence (5′-3′)

IL-1β TGCCACCTTTTGACAGTGATG TGATGTGCTGCTGCGAGATT

IL-6 GACAAAGCCAGAGAGTCCTTCAGA TGTGACTCCAGCTTATCTCTTGG

IL-4 CTCACAGCAACGAAGAACACC CTGCAGCTCCATGAGAACACT

IL-10 GCTGTCATCGATTTCTCCCCT GACACCTTGGTCTTGGAGCTTAT

Mas1 CCTGGCAAAGGCAGGATCTATT CTCCCCTTTTCAATCTTGCGT

## Western blotting assay

Brain tissue samples and cells were lysed with RIPA lysis buffer (Servicebio biological Co., Ltd., Hubei, China) containing 1 mM phenylmethylsulfonyl fluoride on ice for 15 min, followed by centrifugation at 12,000 rpm for 15 min. The supernatant was collected, and the concentrate was quantified by the BCA protein assay kit (Servicebio biological Co., Ltd.,). After denaturation at 100°C for 10 min with an up-sampling buffer, the protein samples were separated on 6% and 10% SDS-PAGE gels at 100 V for 30 min, followed by separation at 150 V for 90 min, and then transferred to polyvinylidene difluoride membranes (Millipore) at 300 mA

for 2 h. The membrane was then blocked with 5% bovine serum albumin (BSA, Servicebio biological Co., Ltd.,) for 1.5 h at room temperature and then incubated with primary antibodies in the blocking buffer at 4°C overnight. On the next day, the membranes were incubated with HRP-conjugated secondary antibodies for 2 h at room temperature. The bands were detected in the dark by using an enhanced chemiluminescence kit (Thermo Fisher, Waltham, MA, USA) in accordance with the manufacturer's protocol. Individual protein bands were quantified by densitometry using ImageJ software (NIH, USA). The following primary antibodies were used in the experiments: anti-CD86 (1:2000, Thermo Fisher Scientific, MA1-10293), anti-CD206 (1:2000, Cell Signaling Technology, #24595), anti-IL-6 (1:2000, Abcam, ab290735), anti-IL-10 (1:2000, Abcam, ab310329), anti-beta Actin (1:2000, Abcam, ab8227), anti-GAPDH (1:2000, Cell Signaling Technology, #2118) and anti-Mas1 (1:1000, Santa Cruz Biotechnology, sc-390453).

## Immunofluorescence assay of tissue samples

Mice were anaesthetized with sodium pentobarbital 1 day after surgery and then perfused through the heart with saline and 4% paraformaldehyde (PFA) at 4°C. Mouse brain tissues were collected and fixed with 4% PFA overnight at 4°C; subsequently, the tissues were placed in 15% and 30% sucrose solutions for dehydration for 3 d. The tissues were embedded in paraffin and frozen at -80°C. The embedded samples were then cut into 10-μm-thick sections. The tissue sections were penetrated and sealed using 0.3% Triton X-100 and 5% BSA for 1 h. Next, the sections were incubated with primary antibodies overnight at 4°C and washed three times with TBST for 10 min each. The sections were then incubated with secondary antibodies in the dark at room temperature for 1 h and washed three times with TBST for 10 min each. Images of each slice were randomly captured using a laser confocal super-resolution microscope, and the fluorescence intensity of each experimental group image was analyzed by ImageJ software. The following primary antibodies were used in the experiments: anti-CD86 (1:500, Cell Signaling Technology, #19589S), anti-CD206 (1:500, Cell Signaling Technology, #24595), and anti-Iba-1 (1:100, Abcam, ab178846).

## Hematoxylin-eosin staining and Nissl staining

Mouse brains were carefully removed 24 h after establishing the model. The brain tissue was fixed with a 4% PFA solution and then embedded in paraffin. Several 4-μm-thick paraffin sections were then prepared. The sections were then subjected hematoxylin-eosin (HE) staining and Nissl staining and observed under a light microscope. In the sections stained by Nissl staining, the number of surviving neurons was counted in 5 randomly selected regions of interest (ROIs).

## Immunohistochemical assay

Brain tissue samples were fixed with formaldehyde, embedded in paraffin, cut into 4-μm-thick sections, rehydrated in gradient alcohol, and incubated with antibodies against the ionized calcium-binding adaptor molecule Iba-1 (1:500; Abcam, Cambridge, UK). The stained sections were then washed and incubated with secondary antibodies for 1 h at room temperature. After the sections were washed with PBS, a DAB chromogenic solution was added for 5 min, and the sections were re-stained with HE. The sections were then subjected to gradient ethanol dehydration and neutral blocking, and three sections per mouse were used for quantitative analysis. The number of Iba-1-positive cells was calculated using ImageJ software.

## Behavioral assessment

The modified neurological severity score (mNSS) and Garcia score were used to detect neurological impairments in mice [14]. In the mNSS test, the higher the score, the worse is the neurological function. In the Garcia test with a score range of 3–21, the higher the score, the better is the neurological function. All tests were assessed using a double-blind method.

The open field test was used to assess the motor and exploratory abilities of mice [15]. Each mouse was placed in the center of a wilderness field equally divided into 16 squares ($40 \times 40 \times 40$ cm) and allowed to move freely for 5 min. A computerized video tracking system was used to record the behavior of the animals. The numbers of crossings and rearings were recorded. The crossing number was defined as the number of times the entire body of mouse crossed the line. The rearing number was defined as the number of times the body was turned upward and both forelimbs were in the air. The next set of tests was performed after the environmental chamber was cleaned. The environment was kept quiet throughout the procedure.

## Statistical analyses

Two-by-two comparisons between three and more groups were analyzed by one-way ANOVA and Tukey's post hoc test, and the results were expressed as mean ± standard deviation. Between-group differences in both scoring tests were analyzed using repeated-measures two-way mixed ANOVA followed by Bonferroni test. For all analyses, $p < 0.05$ was considered statistically significant. SPSS 25.0 (SPSS Inc., Chicago, IL, USA) was used to analyze the data as described previously.

## Result

### Ang-(1–7) induces MasR expression in BV2 cells and affects the release of inflammatory factors under low-temperature-induced condition

BV2 cells were induced by LPS and incubated at 15˚C to establish the LPS+low-temperature-induced (TS) model. To investigate the role of the Ang-(1–7)/MasR axis in BV2 cells under the LPS+low-temperature-induced environment, we conducted qPCR and western blotting (WB) assay to observe the changes in the expression of MasR and inflammatory factors and used A779 to confirm whether the action of Ang-(1–7) was mediated by the MasR pathway. The qPCR assay was used to observe the release of inflammatory factors, and the results showed that the LPS+low-temperature-induced condition upregulated the expression of proinflammatory cytokines in BV2 cells. After further Ang-(1–7) induction, the expression levels of IL-1β and IL-6 were significantly downregulated in the cells, while the expression levels of IL-4 and IL-10 were significantly upregulated; these changes were blocked by A779 (Fig 1A–1D). To further validate this phenomenon, we conducted WB assay to investigate the expression of MasR in TS and TS+Ang-(1–7)+A779-induced cells and to confirm the expression of inflammatory factors. Consistent with the qPCR results, IL-6 expression was downregulated in the TS+Ang-(1–7) group compared to the TS group. Moreover, compared to the control group, IL-10 and MasR expression levels were upregulated in both the TS and TS+Ang-(1–7) groups; in contrast, IL-10 expression levels in the TS+Ang-(1–7) group was higher than that in the TS group (Fig 1E–1H). These results indicate that the expression of anti-inflammatory factors and MasR increased after Ang-(1–7) treatment, while the expression of proinflammatory factors decreased after treatment, thus suggesting that Ang-(1–7) modulates the release of inflammatory factors in MasR-expressing BV2 cells under the low-temperature-induced condition.

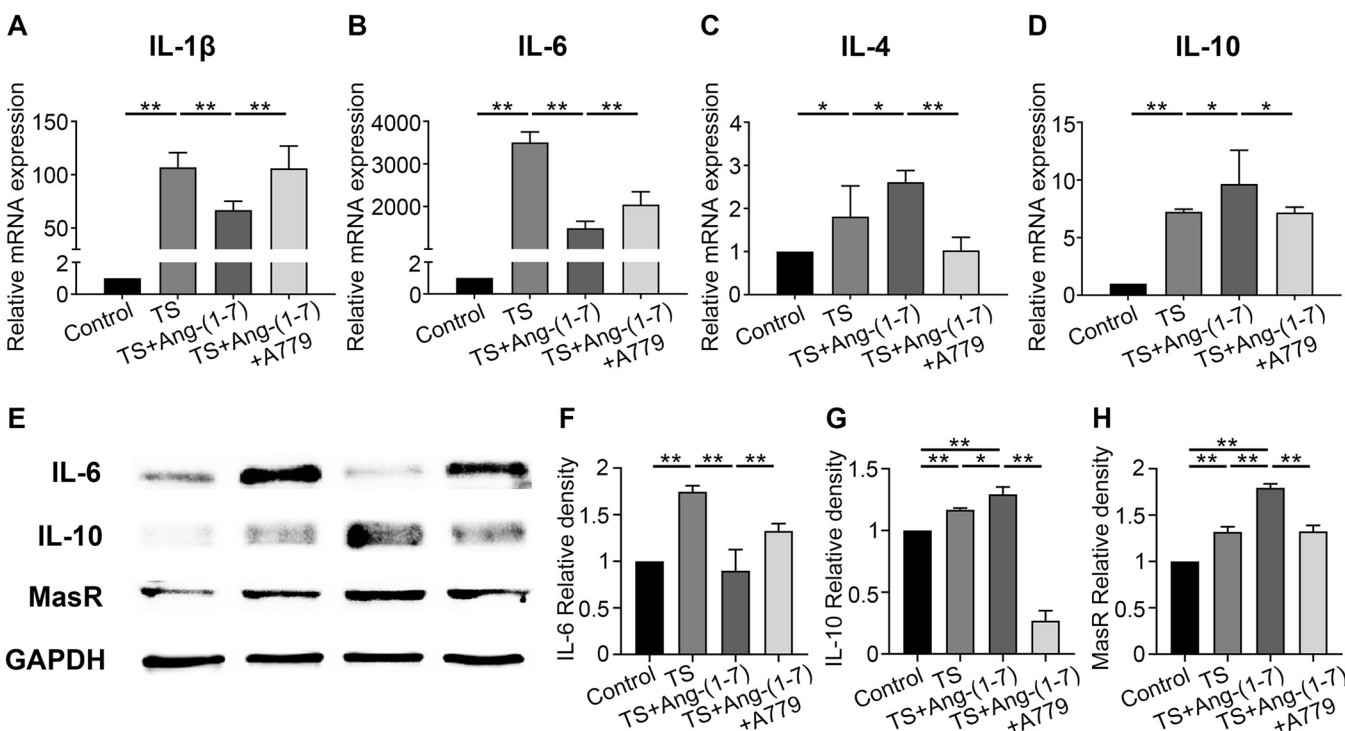

**Fig 1. Ang-(1–7) regulates the transformation of BV2 cells toward the suppression of inflammation.** (A-D) Quantitative RT-qPCR analysis of BV2 cells showing mRNA expression of IL-1β, IL-6, IL-4, and IL-10 after processing. Data are expressed as fold change compared to the control group (n = 3). (E-H) Western blotting assay and quantification showing protein expression of IL-6 and IL-10 in microglia after processing. Data are expressed as fold change compared to the control group (n = 3). P > 0.05; *P < 0.05, **P < 0.01.

## Ang-(1–7)/MasR modulates microglia polarization in the brain cortex of TS mice

In the *in vivo* experiment, we assessed the polarizing effect of Ang-(1–7) on microglia in TS mice by immunofluorescence assay. CD86 and CD206 were used as markers for M1 and M2 microglia, respectively. The results showed that the expression levels of CD86 and CD206 in the brain cortex of the TS group were significantly higher than those in the control group. And compared with the TS group, the CD86 expression level was significantly decreased in the brain cortex of TS mice treated with Ang-(1–7),in contrast, the CD206 expression level was significantly increased. Additionally, A779 acted as a MasR antagonist to reverse the polarizing effect of Ang-(1–7) (Fig 2A–2C). This trend was further confirmed by WB assay. Similar to the results of the immunofluorescence assay, the microglia marker and cytokine expression levels were higher in the TS group than in the control group. Compared to the TS group, the TS+Ang-(1–7) group showed decreased expression of CD86 and IL-6 and increased expression of CD206 and IL-10 (Fig 2D–2H). Additionally, MasR expression was higher in the TS and TS+Ang-(1–7) groups than in the control group; however, MasR expression was lower in the TS+Ang-(1–7)+A779 group than in the TS+Ang-(1–7) group (Fig 2I). Taken together, Ang-(1–7) could transform microglia phenotype from M1 to M2 by binding to MasR.

## Ang-(1–7)/MasR attenuates neuroinflammation and damage in TS mice

The degree of damage and inflammatory cell infiltration in the cerebral cortex of mice was assessed by staining embedded tissue sections. The results showed that the control group mice

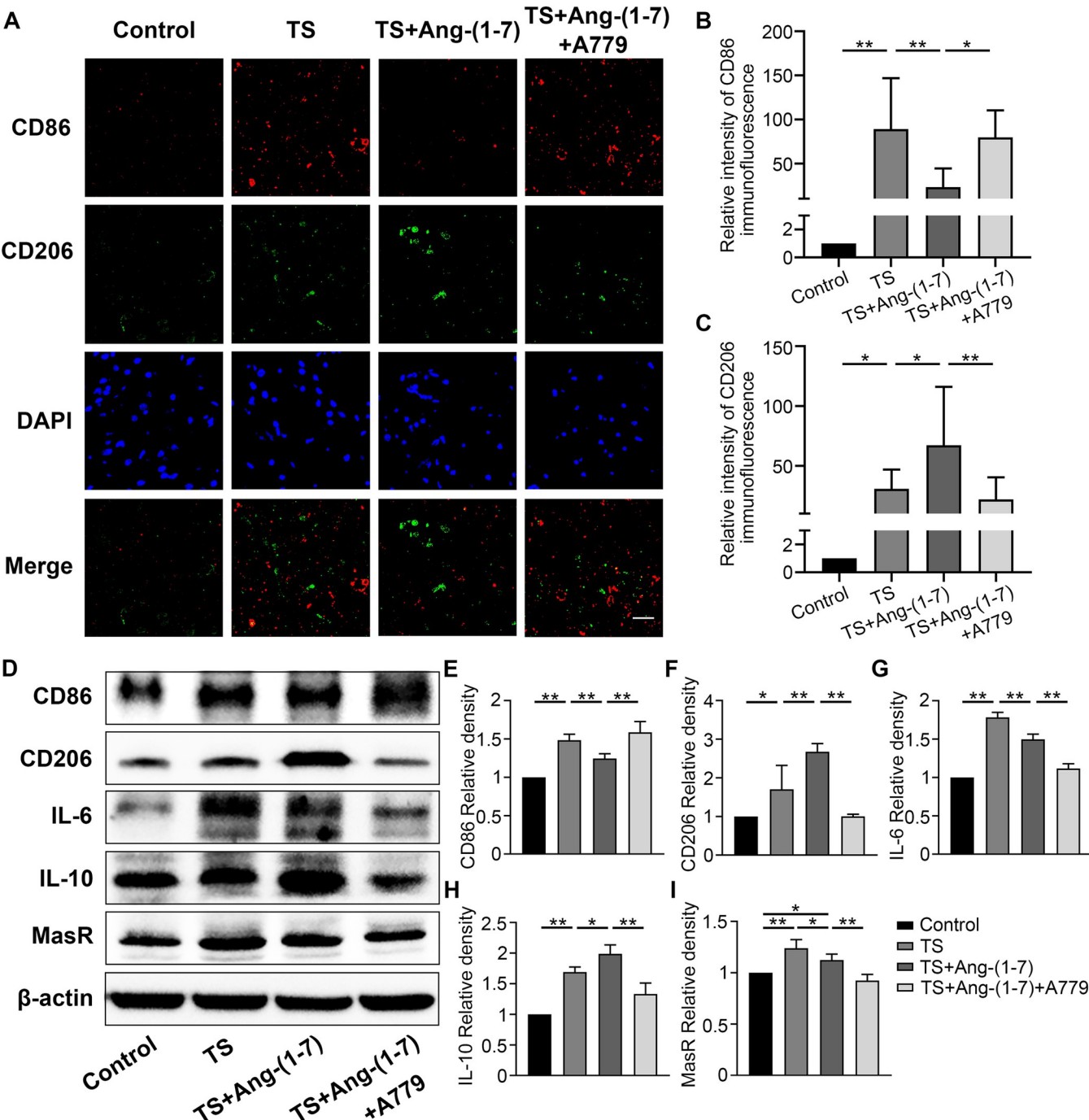

**Fig 2. Ang-(1–7) regulates microglia polarization toward the M2 phenotype.** (A-C) Double staining of CD86 (red)/CD206 (green) in microglia of each group. Data are expressed as fold change compared to the control group (scale bar: 40 μm, n = 6). (D-I) Western blotting assay and quantitative expression of markers, cytokines, and MasR in microglia of each group. Data are expressed as fold change compared to the control group (n = 3). P > 0.05; *P < 0.05, **P < 0.01.

did not exhibit hemorrhage or brain edema, and their tissue structure was intact. In contrast, scattered hemorrhages were found in the TS group mice, with deformation of tissue structure and cytoarchitecture and aggregation of erythrocytes in and around the area of impact. The TS

+Ang-(1–7) group showed slightly less tissue destruction, with partial reconstruction of small blood vessels and more confined inflammation (Fig 3A). Nissl staining was used to further characterize the neuronal damage. Compared to the control group, the TS group showed neuronal cell deficits and disorders and reduced or even absence of cellular Nissl vesicles. In contrast, the TS+Ang-(1–7) group showed an increased proportion of normal neurons (Fig 3B). Immunohistochemical staining of Iba-1 was used to determine the degree of inflammatory cell infiltration around the damaged cerebral cortex. The results showed that the Iba-1 positivity rate in the TS group was significantly higher than that in the control group, while the Iba-1 positivity rate around the damaged area in the Ang-(1–7) treatment group was significantly lower than that in the TS group. Additionally, the TS+Ang-(1–7)+A779 group showed reversal of the effect of Ang-(1–7) and increased positive expression of Iba-1 (Fig 3C). These results indicated that Ang-(1–7) participates in neuronal repair and attenuates neuroinflammation through the MasR pathway.

## Ang(1–7) improves the recovery of behavioral function after injury in TS mice

The role of the Ang-(1–7)/MasR axis in TS mice was confirmed by an open field experiment and two neurological function scoring systems. The open-field experiment showed that throughout the recovery period within 14 d after model establishment, the TS+Ang-(1–7) group exhibited a rapid increase in the numbers of crossing and rearing, with better recovery of motor function and exploratory ability; in contrast, the TS+Ang-(1–7)+A779 group showed slow recovery of motor function and exploratory ability as compared to the TS+Ang-(1–7) group (Fig 4A–4C). The results of both mNSS and Garcia scores indicated that TS mice treated with Ang-(1–7) showed different degrees of improvement in neurological function, and this therapeutic effect was inhibited by A779 (Fig 4D and 4E). In conclusion, the Ang-(1–7)/MasR axis improves functional recovery in TS mice, and the MasR pathway plays an important role in mediating the effect of Ang-(1–7).

## Discussion

Neuroinflammation is an acute response that occurs after TBI and is characterized by the migration of resident cells to the injury site, followed by release of inflammatory factors [16]. This response is mediated by two pathways. In the first pathway, microglia play a neuroprotective role immediately after injury by phagocytizing and removing damaged cell debris and releasing anti-inflammatory cytokines and neurotrophic factors. The proportion of M2-type microglia gradually increases in the week after injury, leading to tissue remodeling and inhibition of the inflammatory response [17–19]. In the second pathway, M1-type microglia release proinflammatory cytokines and oxidative metabolites in the initial period, thereby promoting inflammation [20,21]. Neuroinflammatory modulation is considered an important strategy to repair TBI by attenuating secondary brain damage. In the present study, we modulated the M1/M2 polarization of microglia through Ang-(1–7)/MasR axis activation to treat more severe neuroinflammation that occurs in hypothermic disturbances and to improve histological and functional outcomes in TBI-SIH mice (Fig 5). The MasR antagonist A779 significantly inhibited the polarization of microglia toward the M2 type, which exacerbated secondary brain damage and dysfunction after TBI-SIH.

Ang-(1–7) is a critical neuromodulator, particularly for the brain and cardiovascular system [22,23]. Several studies have confirmed that Ang-(1–7) is a key molecule in inflammation regulation [24–26]. Following inflammatory exposure, MasR is upregulated in macrophages, and Ang-(1–7) regulates the expression levels of proinflammatory cytokines IL-6 and TNF-α and

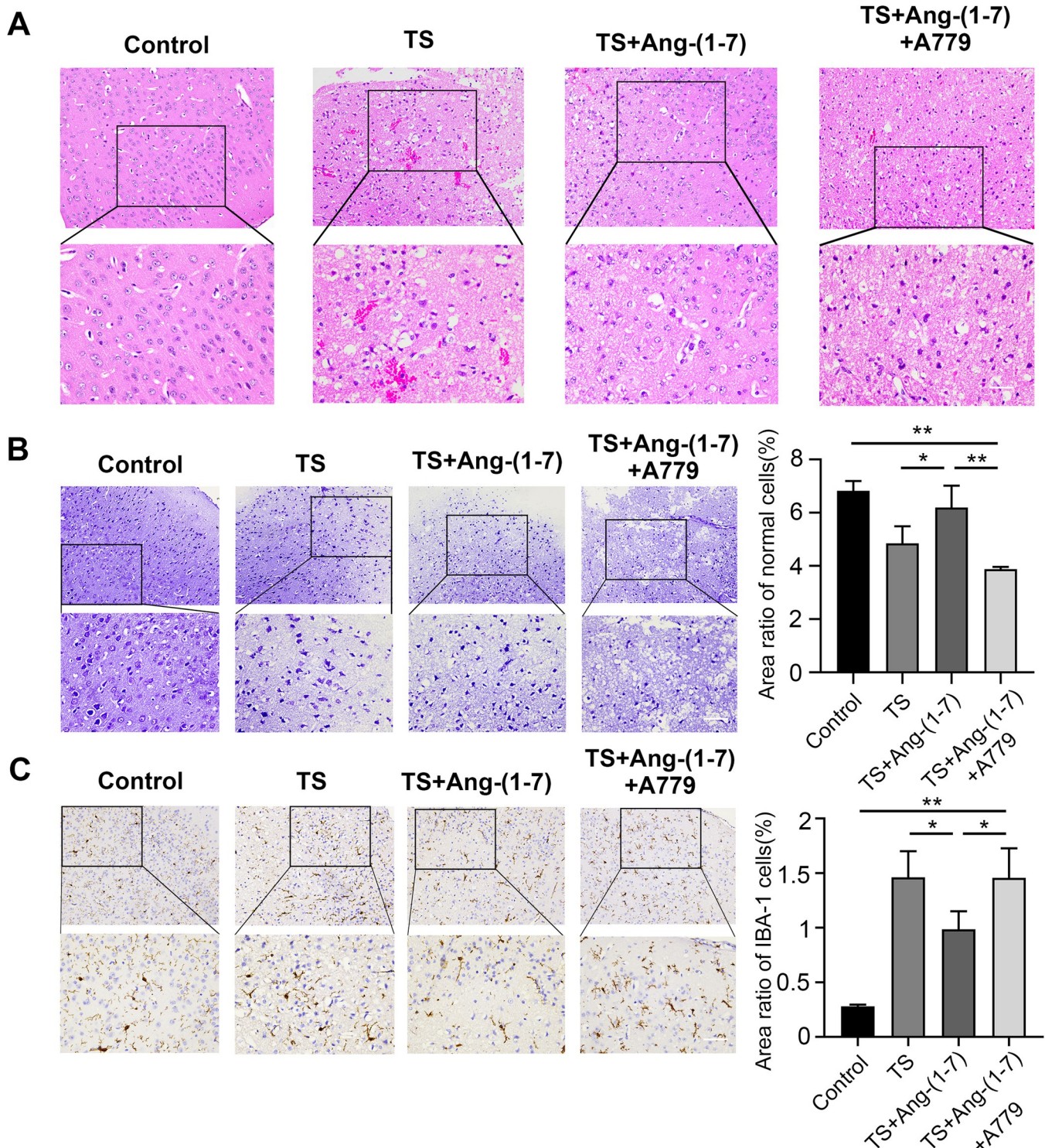

**Fig 3. Histological results of brain tissues from Ang-(1–7)-treated TS mice.** (A) HE staining at 24 h after modeling (scale bar: 40 μm, n = 3). (B) Nissl staining and the proportion of regional normal neurons (scale bar: 40 μm, n = 3). (C) Immunohistochemical assay and the proportion of Iba-1-positive cells in the region (scale bar: 40 μm, n = 3). *P < 0.05, **P < 0.01.

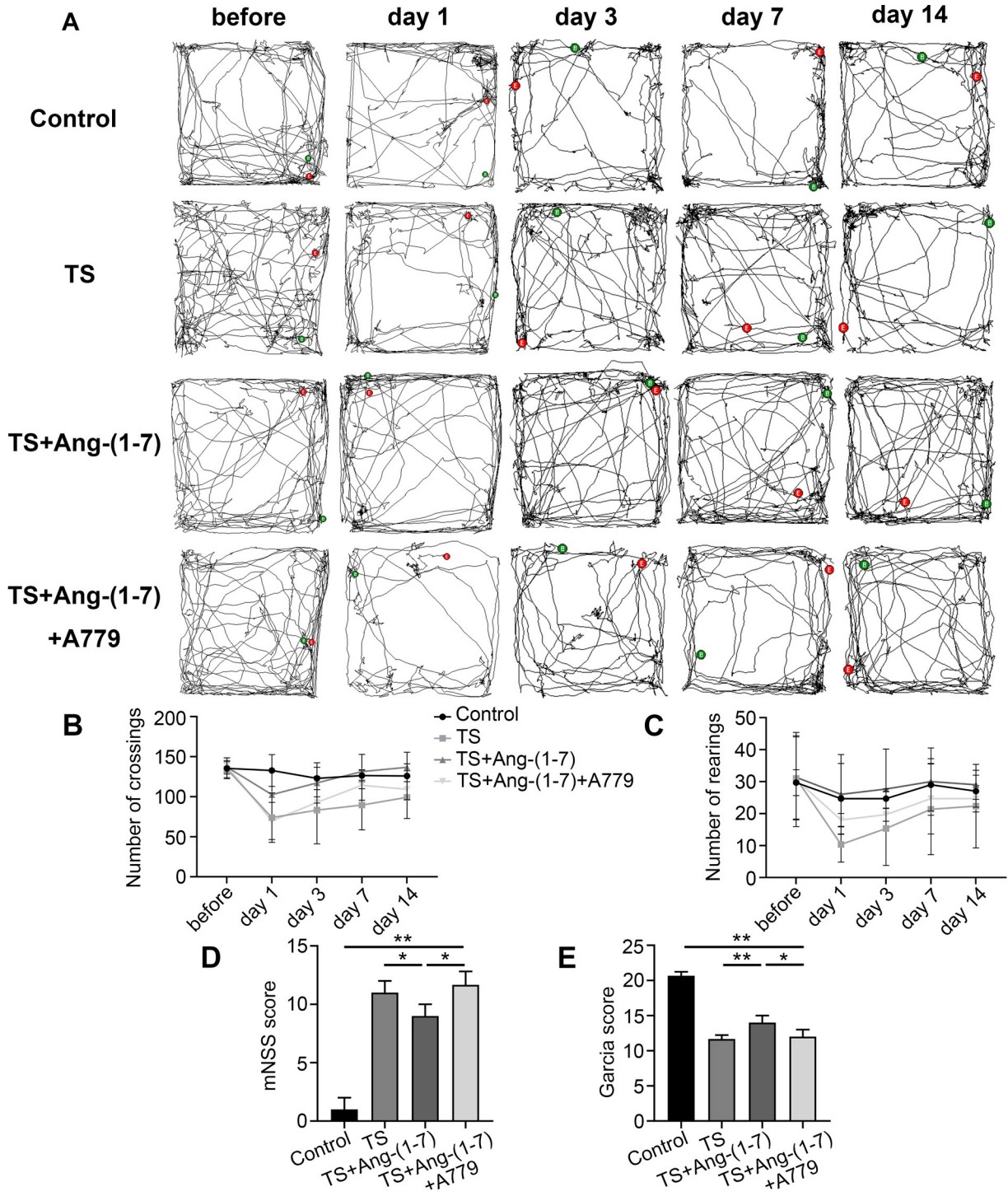

**Fig 4. Ang-(1–7) improved motor function in TS mice.** (A) Open field experiments of locomotion pattern in mice. (B-C) Crossing and rearing numbers in mice (n = 3). (D) mNSS score (n = 3). (E) Garcia score (n = 3). *P < 0.05, **P < 0.01.

upregulates IL-10 expression levels [24,27]. Therefore, we attempted to determine whether Ang-(1–7) has an inhibitory effect on inflammation in hypothermic TBI. The immunofluorescence assay of mouse brain tissues showed that the expression of CD206 and CD86 was

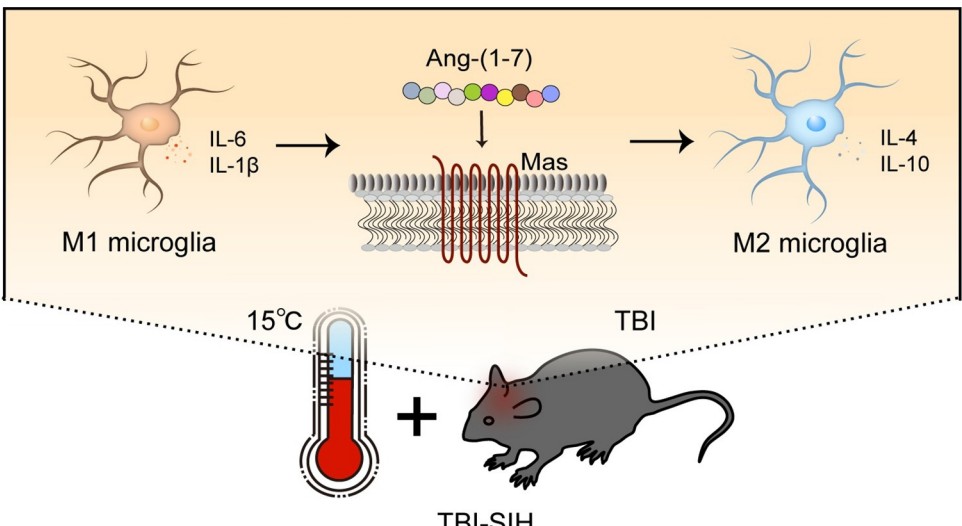

**Fig 5. Schematic illustration of Ang-(1–7)/MasR axis regulating M1 microglia polarisation towards M2 in the brain of hypothermic traumatic brain injury mice.**

upregulated and downregulated, respectively, in infiltrating microglia as compared to that in the TS group. The results of WB assay further confirmed that IL-6 expression was decreased and IL-10 expression was increased. These results suggest that Ang-(1–7) can regulate microglia polarization toward the M2 type *in vivo*, which is consistent with the results of the *in vitro* experiment. In addition, MasR expression was also upregulated in the TS group, suggesting that the upregulation of MasR may be related to the macrophage activation status rather than to a specific phenotype [13]. Interestingly, HE staining revealed a more pronounced demarcation between the damaged area and the surrounding normal tissue in the TS+Ang-(1–7) group than in the TS group. Ang-(1–7) may play a role in rapidly recruiting macrophages to encapsulate the inflammation site for limiting the progression of inflammation [28]. The results of the immunohistochemical assay showed a decrease in the positivity of the microglia marker Iba-1 in the TS+Ang-(1–7) group and a rebound of Iba-1 positivity following the addition of A779; this may be related to the migration and phenotypic transformation of microglia in the acute phase of injury [29].

Open field experiments and neurological function scores are frequently used for the prognostic analysis of various types of neurological injuries. Our results showed that during the 14-day functional recovery phase, the Ang-(1–7) group mice had more crossing and rearing numbers than the TS group mice, thus implying that both motor function and exploratory ability were restored to a certain extent in the Ang-(1–7) group. At 24 h after surgery, the neurological function scores of the Ang-(1–7) group were better than those of the TS group. These results suggest that Ang-(1–7) can effectively alleviate hypothermic TBI and promote the recovery of histological and motor functions in mice. This might be related to the property of Ang-(1–7) to inhibit the expression of proinflammatory factors, which is consistent with the findings of Jiang T. [30]. Our study confirmed that Ang(1–7) plays a neuroprotective role in hypothermic TBI. The selective antagonist A779 blocks MasR, the specific receptor of Ang-(1–7) [31]. Interestingly, in all our experiments, the effects of Ang-(1–7) were inhibited by the antagonistic effect of A779 on MasR, thus confirming that the function of Ang-(1–7) *in vivo* is mediated by the MasR receptor. In addition to the WB results for IL-6, there may be other pathways in this that interfere with cytokine expression.

The present study has some limitations. First, we only focused on determining the effects of Ang-(1–7) in mice *in vitro* and *in vivo*, with no strategy for clinical application. Second, neuroinflammation after TBI progresses through a complex process, and the mechanism of action of hypothermia and Ang-(1–7) *in vivo* and their mutual interaction require further studies.

## Conclusion

In conclusion, the present study revealed a regulatory role of the Ang-(1–7)/MasR axis in hypothermic neuroinflammation, wherein it ameliorated the inflammatory milieu by altering the polarization of the M1/M2 phenotype of microglia. The development of neuroinflammation is also closely associated with disease prognosis, and this study provides a new direction for treating hypothermic TBI and promoting functional rehabilitation of patients with TBI-SIH.

## Supporting information

**S1 Raw images.**
(TIF)

**S1 Raw data.**
(DOCX)

## Author Contributions

**Conceptualization:** Dan Ye, Jiamin Liu, Long Lin.

**Funding acquisition:** Shousen Wang.

**Methodology:** Dan Ye, Jiamin Liu, Long Lin.

**Supervision:** Dan Ye, Long Lin.

**Visualization:** Dan Ye, Jiamin Liu.

**Writing – original draft:** Dan Ye.

**Writing – review & editing:** Pengwei Hou, Tianshun Feng, Shousen Wang.

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
