## [Decision Letter · Decision Letter 0]

9 Jan 2024

PONE-D-23-41383The Ang-(1-7)/MasR axis ameliorates neuroinflammation in hypothermic traumatic brain injury in mice by modulating phenotypic transformation of microgliaPLOS ONE

Dear Dr. Wang,

Thank you for submitting your manuscript to PLOS ONE. After careful consideration, we feel that it has merit but does not fully meet PLOS ONE’s publication criteria as it currently stands. Therefore, we invite you to submit a revised version of the manuscript that addresses the points raised during the review process. Please submit your revised manuscript by Feb 23 2024 11:59PM. If you will need more time than this to complete your revisions, please reply to this message or contact the journal office at plosone@plos.org. Please include the following items when submitting your revised manuscript:A rebuttal letter that responds to each point raised by the academic editor and reviewer(s). You should upload this letter as a separate file labeled 'Response to Reviewers'.A marked-up copy of your manuscript that highlights changes made to the original version. You should upload this as a separate file labeled 'Revised Manuscript with Track Changes'.An unmarked version of your revised paper without tracked changes. You should upload this as a separate file labeled 'Manuscript'.

We look forward to receiving your revised manuscript.

Kind regards,

Michael Bader

Academic Editor

PLOS ONE

Journal Requirements:

5. We note that your Data Availability Statement is currently as follows: [All relevant data are within the manuscript and its Supporting Information files.]

7. Please remove your figures from within your manuscript file, leaving only the individual TIFF/EPS image files, uploaded separately. These will be automatically included in the reviewers’ PDF.

8. PLOS ONE now requires that authors provide the original uncropped and unadjusted images underlying all blot or gel results reported in a submission’s figures or Supporting Information files. This policy and the journal’s other requirements for blot/gel reporting and figure preparation are described in detail at https://journals.plos.org/plosone/s/figures#loc-blot-and-gel-reporting-requirements and https://journals.plos.org/plosone/s/figures#loc-preparing-figures-from-image-files. When you submit your revised manuscript, please ensure that your figures adhere fully to these guidelines and provide the original underlying images for all blot or gel data reported in your submission. See the following link for instructions on providing the original image data: https://journals.plos.org/plosone/s/figures#loc-original-images-for-blots-and-gels.

Reviewers' comments:

Reviewer's Responses to Questions

**Comments to the Author**

1. Is the manuscript technically sound, and do the data support the conclusions?

Reviewer #1: Partly

Reviewer #2: Partly

2. Has the statistical analysis been performed appropriately and rigorously? 

Reviewer #1: N/A

Reviewer #2: Yes

3. Have the authors made all data underlying the findings in their manuscript fully available?

Reviewer #1: Yes

Reviewer #2: Yes

4. Is the manuscript presented in an intelligible fashion and written in standard English?

Reviewer #1: Yes

Reviewer #2: Yes

5. Review Comments to the Author

Reviewer #1: The study by Ye and colleagues aimed to determine whether the Ang-(1-7)/MasR axis exerts a therapeutic effect on hypothermic TBI and the underlying mechanisms; authors believe that the findings of their study could guide to develop appropriate treatment methods for TBI-SIH. Authors found that hypothermia exacerbates TBIinduced damage and that the Ang-(1-7)/MasR axis can ameliorate hypothermic TBI and directly affect prognosis.

The results are someway interesting; however, the manuscript needs major improvement; 1) the rationale and hypothesis seems preposterous and, 2) the BV2 cells protocol detract focus from the manuscript. I suggest focusing on TBI combined with 15°C seawater immersion (TS) cells data only.

Abstract:

Needs improvement.

First sentence; evidence suggests or indicates or strongly indicates that the Ang-(1-7)/MasR axis may contribute for the treatment of several diseases

The following statement is quite preposterous… “…The Ang-(1-7)/MasR axis is critically involved in treating several diseases; however, its function in traumatic brain injury (TBI) combined with seawater immersion hypothermia remains unclear (??)…” many mechanisms must be unclear under this condition…what happens with Ang II in this condition?? It is not a casual and of course there is a lot beyond your question that remains to be discovered in these peculiar conditions…However, if this reviewer is wrong, please, include references for other RAS components or peptides in the same condition for comparison purposes; otherwise, the main question and hypothesis of the ms. needs to be reformulated. Perhaps, this statement could only be used briefly at the end of discussion.

Introduction:

See above: Please, recommend complete modification of the following statement: “…However, the

pathophysiological mechanisms associated with the development of TBI combined

with seawater immersion hypothermia (TBI-SIH) remain unclear, and currently…”

Methods

Relevance of the BV2 cell culture and model establishment in the context of ms needs clarification. The BV2 cells protocol detract focus from the manuscript. I suggest focusing on TS cells data only.

From a translational perspective, as the author emphasizes, hypothermic seawater + TBI and a BV2 cell model of hypothermic inflammation are absolutely different approaches.

Reviewer #2: The authors show interesting data to the field.

I have some comments:

The in vitro model includes LPS and exposure of cells to low temperature. They also should show cells' response to each one alone.

Fig 1D, It does not look like A779 antagonized cells' response to TS+Ang 1-7. Please check. Fig 1 F,G- antagonizing effects of A779 are not seen. Please explain.

Primary microglia should be used at least for part of the study to make sure that data with cell line are representative.

Fig 2. - Pictures are not clear.

6. PLOS authors have the option to publish the peer review history of their article (what does this mean?). If published, this will include your full peer review and any attached files.

Reviewer #1: No

Reviewer #2: No

---

## [Author Response · Author response to Decision Letter 0]

23 Mar 2024

Dear Editor,

Thank you very much for your replies and all reviewers’ kindly comments of our manuscript entitled “The Ang-(1-7)/MasR axis ameliorates neuroinflammation in hypothermic traumatic brain injury in mice by modulating phenotypic transformation of microglia” (PONE-D-23-41383). We have carefully considered these comments, and hereby submit our point-by-point responses to your and the Reviewers’ comments, editorial corrections in our revised manuscript.

We are pleased that the reviewers believe that our work is meaningful and contributes to the fields of hypothermia and neurotrauma. Considering your and the reviewer's comments, we have made revisions to several parts of the manuscript. These comments are very valuable for revising and improving our manuscript, and also provide good ideas for our current and future research. Based on the suggestions of the reviewers, we have revised the abstract and introduction sections, and added annotations on issues such as individual factor interventions in the in vitro model, unclear antagonistic effects of A779, and limitations in using BV2 cells. We believe that such modifications can make the content of our manuscript more complete and interesting. In addition, we have improved the manuscript and corrected all errors in the revised manuscript according to the journal requirements.

We are very grateful to you and the reviewers for their help in improving this article.

Yours sincerely,

Shousen Wang, MD, PhD

Reviewers' Comments:

General response: We would like to thank the reviewers for their time and insightful comments. We have carefully considered these comments and substantially revised the manuscript by thoroughly researching the field reviewed and clarifying the content of this manuscript to make it more comprehensive. Below, we address each of the reviewers’ questions.

Reviewer 1

1. the rationale and hypothesis seems preposterous

Our response: Thanks very much for your constructive suggestions. The TS model is a model we proposed to simulate the state of hypothermia combined with traumatic brain injury, and in our previous study, we found that hypothermia exacerbates neuroinflammation after traumatic brain injury[1]. As well as in this paper, the cytokines and markers related to neuroinflammation in the TS group were significantly higher than those in the control group , and Ang-(1-7) was used to attenuate neuroinflammation after traumatic brain injury[2], based on this experimental basis we proposed the hypothesis of this paper: Ang-(1-7) attenuates the neuroinflammation caused by TS and this response is antagonised by A779. We acknowledge the limitations of this hypothesis, and we will take the reviewers' comments into account in subsequent studies to further explore the relevant mechanisms to refine this hypothesis. In addition, we apologise for the lack of clarity in our presentation and have made changes in the abstract and introduction in response to your suggestions.

1.Dan Ye, Long Lin, Pengwei Hou, et al. Establishment and evaluation of a mouse model of traumatic brain injury combined with seawater immersion induced hypothermia. Chinese Journal of Naval Medicine and Hyperbaric Medicine, 2023;30 (6):697-702. doi: 10.3760/cma.j.cn311847-20230711-00173.

2.Bruhns RP, Sulaiman MI, Gaub M, et al. Angiotensin-(1-7) improves cognitive function and reduces inflammation in mice following mild traumatic brain injury. Front Behav Neurosci. 2022;16:903980. doi: 10.3389/fnbeh.2022.903980.

2.the BV2 cells protocol detract focus from the manuscript. I suggest focusing on TBI combined with 15℃ seawater immersion (TS) cells data only.

Our response: Thank you for carefully reviewing our manuscript and giving advice.We acknowledge that experiments using TS cells can prove the experimental conclusions on the other hand. However, our study mainly explores the transformation of microglia under TS conditions in response to Ang-(1-7) and the role this transformation plays in neuroinflammation. The use of LPS to stimulate BV2 cells to mimic the pro-inflammatory response of microglia after TBI is still widely accepted in neuroinflammatory studies[1, 2]. Our previous studies have also shown that stimulation of BV2 cells using LPS can produce an inflammatory response similar to that of microglia after TBI. That is why we applied it in this experiment. Of course the advice you gave is very important and we will consider using TS cells for the next experiments afterwards.

1.Henn A, Lund S, Hedtjärn M, et al. The suitability of BV2 cells as alternative model system for primary microglia cultures or for animal experiments examining brain inflammation. Altex. 2009;26(2):83-94. doi: 10.14573/altex.2009.2.83.

2.Hua T, Yang M, Song H, et al. Huc-MSCs-derived exosomes attenuate inflammatory pain by regulating microglia pyroptosis and autophagy via the miR-146a-5p/TRAF6 axis. J Nanobiotechnology. 2022;20(1):324. doi: 10.1186/s12951-022-01522-6.

3.Abstract:Needs improvement.

First sentence; evidence suggests or indicates or strongly indicates that the Ang-(1-7)/MasR axis may contribute for the treatment of several diseases. The following statement is quite preposterous;The Ang-(1-7)/MasR axis is critically involved in treating several diseases; however, its function in traumatic brain injury (TBI) combined with seawater immersion hypothermia remains unclear; many mechanisms must be unclear under this condition; what happens with Ang II in this condition?? It is not a casual and of course there is a lot beyond your question that remains to be discovered in these peculiar conditions; However, if this reviewer is wrong, please, include references for other RAS components or peptides in the same condition for comparison purposes; otherwise, the main question and hypothesis of the ms. needs to be reformulated. Perhaps, this statement could only be used briefly at the end of discussion.

Our response: We are very grateful for the reviewer to read our articles carefully.We apologise that our misrepresentation has diverted the sentence from its intended meaning.Ang-(1-7) has been shown to reduce inflammation and improve neurological function in spinal cord injuries and traumatic brain injuries[1, 2], and Ang-(1-7) has an inhibitory effect on post-inflammatory hypothermia[3]. With this in mind, what role Ang-(1-7) has in terms of neuroinflammation and neurological function in traumatic brain injury combined with hypothermia is not yet known to us. Thank you very much for your critical questions, and based on your suggestions, we have revised the abstract to make it more rigorous.

1.Gu G, Zhu B, Ren J, et al. Ang-(1-7)/MasR axis promotes functional recovery after spinal cord injury by regulating microglia/macrophage polarization. Cell Biosci. 2023;13(1):23. doi: 10.1186/s13578-023-00967-y.

2.Bruhns RP, Sulaiman MI, Gaub M, et al. Angiotensin-(1-7) improves cognitive function and reduces inflammation in mice following mild traumatic brain injury. Front Behav Neurosci. 2022;16:903980. doi: 10.3389/fnbeh.2022.903980.

3.Souza LL, Duchene J, Todiras M, et al. Receptor MAS protects mice against hypothermia and mortality induced by endotoxemia. Shock. 2014;41(4):331-336. doi:10.1097/SHK.0000000000000115.

4.Introduction: See above: Please, recommend complete modification of the following statement: However, the pathophysiological mechanisms associated with the development of TBI combined with seawater immersion hypothermia (TBI-SIH) remain unclear, and currently.

Our response: We apologize for not expressing ourselves clearly enough and misleading you. Based on your suggestions, we have revised the inappropriate descriptions in the introduction, and we have modified the original semantics and order of words to make the context more closely related.

5.Methods: Relevance of the BV2 cell culture and model establishment in the context of ms needs clarification. The BV2 cells protocol detract focus from the manuscript. I suggest focusing on TS cells data only.From a translational perspective, as the author emphasizes, hypothermic seawater + TBI and a BV2 cell model of hypothermic inflammation are absolutely different approaches.

Our response: Thank you for your valuable suggestion and let us recognize the shortcomings existing in our study.Using LPS to stimulate BV2 cells to simulate the pro-inflammatory response of microglia after TBI was the aim of our study using BV2 cells. We wanted to validate our results from multiple perspectives, both in vivo and in vitro. We used BV2 cell responses under hypothermic inflammatory conditions to simulate the transformation of microglia in mice brain tissue under TS conditions. As stated previously, we focused on studying the response of microglia after TS, and stimulation of BV2 cells can produce an inflammatory response similar to that of microglia after TBI, so we used the in vitro experimental scheme of LPS + low-temperature induction to achieve similar results as shown on the in vivo experiments. From a translational point of view, such a scheme makes the in vitro experiments more stable and easier to replicate. We will take your suggestion into consideration and explore TS cells in our future research.

Reviewer 2

1.The in vitro model includes LPS and exposure of cells to low temperature. They also should show cells response to each one alone.

Our response:We agree with your sincere suggestion.Therefore, We used LPS and low temperature as separate intervening factors for grouping and re-validated the effect of each factor alone on BV2 cells, and found that LPS+low temperature promoted the release of inflammatory factors and the release of inflammatory factors was more obvious than that of the LPS group, which meant that the low temperature was likely to aggravate the inflammatory response induced by LPS in vitro, which was exactly the same as the results of our previous in vivo experiments. Interestingly, the expression of MasR showed the same trend, which may imply that hypothermia affects the Ang-(1-7)/MasR axis to a certain extent, and the exact mechanism of the effect needs to be further evidenced. Thank you for providing us with a new direction for our next research programme.（Original images are provided in the supporting materials.）

2.Fig 1D, It does not look like A779 antagonized cells response to TS+Ang 1-7. Please check. Fig 1 F,G- antagonizing effects of A779 are not seen. Please explain.

Our response: Thanks for your serious attitude.We are sorry that the selected images do not illustrate our results well, we have replaced the Fig F,G corresponding protein blotting graphs with clearer ones and redone the statistical analysis according to your suggestion.The antagonistic effect of A779 on the pro-inflammatory cytokine IL-6 is shown by the higher release of IL-6 in the A779 intervention group than in the Ang-(1-7) treated group, and the antagonistic effect on IL-10 is shown in the release of the inflammation-inhibiting cytokine IL-10 was less in the A779 group than in the Ang-(1-7) treated group, so the antagonistic effect of A779 in Figure 1D is correct. Images of relevant experiments are provided in the Supporting Materials.

3.Primary microglia should be used at least for part of the study to make sure that data with cell line are representative.

Our response: Thanks for your constructive comment.The use of LPS to stimulate BV2 cells to simulate the pro-inflammatory response of microglia after TBI is still widely accepted in neuroinflammatory studies[1, 2]. Our previous studies have also shown that stimulation of BV2 cells using LPS can produce an inflammatory response similar to that of microglia after TBI. Therefore, the use of BV2 cells is more stable and easily accessible. That is why we applied it in this experiment. We will also be using primary microglia in future experiments based on the suggestions you have given us.

1.Henn A, Lund S, Hedtjärn M, et al. The suitability of BV2 cells as alternative model system for primary microglia cultures or for animal experiments examining brain inflammation. Altex. 2009;26(2):83-94. doi: 10.14573/altex.2009.2.83.

2.Hua T, Yang M, Song H, et al. Huc-MSCs-derived exosomes attenuate inflammatory pain by regulating microglia pyroptosis and autophagy via the miR-146a-5p/TRAF6 axis. J Nanobiotechnology. 2022;20(1):324. doi: 10.1186/s12951-022-01522-6.

4.Fig 2. Pictures are not clear.

Our response: We apologize that the provided images are not clear enough.We have re-adjusted the image resolution and brightness of Fig 2 to present our results more clearly.

We would like to take this opportunity to thank you for all your time involved and this great opportunity for us to improve the manuscript. We hope you will find this revised version satisfactory.

Thank you and best regards.

Yours sincerely.

Shousen Wang, MD, PhD

Department of Neurosurgery, 

900th Hospital, 

Fuzhou 350025, Fujian Province, China.

Tel: + (86)13950482966.

E-mail address: wshsen1965@126.com

---

## [Editor Report · Decision Letter 1]

4 Apr 2024

PONE-D-23-41383R1The Ang-(1-7)/MasR axis ameliorates neuroinflammation in hypothermic traumatic brain injury in mice by modulating phenotypic transformation of microgliaPLOS ONE

Dear Dr. Wang,

Thank you for submitting your manuscript to PLOS ONE. After careful consideration, we feel that it has merit but does not fully meet PLOS ONE’s publication criteria as it currently stands. Therefore, we invite you to submit a revised version of the manuscript that addresses the points raised during the review process. Despite that you have answered most the comments of the reviewers a screen of your original Western blot data shows that most blots show multile bands. How did you select the one you used for quantification? Especially for MasR you select different bands in different blots (compare the blot in your answer to reviewers to the others). Please clarify how you secured the specificity of each antibody. Please submit your revised manuscript by May 19 2024 11:59PM. If you will need more time than this to complete your revisions, please reply to this message or contact the journal office at plosone@plos.org. Please include the following items when submitting your revised manuscript:A rebuttal letter that responds to each point raised by the academic editor and reviewer(s). You should upload this letter as a separate file labeled 'Response to Reviewers'.A marked-up copy of your manuscript that highlights changes made to the original version. You should upload this as a separate file labeled 'Revised Manuscript with Track Changes'.An unmarked version of your revised paper without tracked changes. You should upload this as a separate file labeled 'Manuscript'.

We look forward to receiving your revised manuscript.

Kind regards,

Michael Bader

Academic Editor

PLOS ONE

---

## [Author Response · Author response to Decision Letter 1]

16 Apr 2024

Dear Editor,

Thank you very much for your replies and all reviewers’ kindly comments of our manuscript entitled “The Ang-(1-7)/MasR axis ameliorates neuroinflammation in hypothermic traumatic brain injury in mice by modulating phenotypic transformation of microglia” (PONE-D-23-41383). We apologise that some of the details in the manuscript have confused you. We have carefully considered your comments and supplemented with sources of relevant antibodies in Materials and Methods. We selected the bands by the size of the molecular weight of the samples in the specification, and the selected bands are all located within the example range in the specification. Following are the instructions for the antibodies used in the Western blot.

As for the band of MasR, we apologize for the confusion caused by the inappropriate labelling. In the letter to the reviewers, the MasR we labelled seems to deviate from his own molecular weight size, but a look at the whole figure reveals that the bands in either interval express the same tendency, i.e., that hypothermia affects the Ang-(1-7)/MasR axis to a certain extent. Previously we repeated the experiment and got the same conclusion. We are sorry that the image we chose may not be clear enough, and this more visual figure is attached below to provide you an explanation, which we hope will help to answer your confusion.

We would like to take this opportunity to thank you for all your time involved and this great opportunity for us to improve the manuscript. We hope you will find this revised version satisfactory.

Thank you and best regards.

Yours sincerely.

Shousen Wang, MD, PhD

Department of Neurosurgery, 

900th Hospital, 

Fuzhou 350025, Fujian Province, China.

Tel: + (86)13950482966.

E-mail address: wshsen1965@126.com

---

## [Editor Report · Decision Letter 2]

22 Apr 2024

The Ang-(1-7)/MasR axis ameliorates neuroinflammation in hypothermic traumatic brain injury in mice by modulating phenotypic transformation of microglia

PONE-D-23-41383R2

Dear Dr. Wang,

We’re pleased to inform you that your manuscript has been judged scientifically suitable for publication and will be formally accepted for publication once it meets all outstanding technical requirements.

Kind regards,

Michael Bader

Academic Editor

PLOS ONE
---

## [Editor Report · Acceptance letter]

29 Apr 2024

PONE-D-23-41383R2 

PLOS ONE

Dear Dr. Wang, 

I'm pleased to inform you that your manuscript has been deemed suitable for publication in PLOS ONE. Congratulations! Your manuscript is now being handed over to our production team.

Kind regards, 

on behalf of

Prof. Michael Bader 

Academic Editor

PLOS ONE